# A Machine Learning Specklegram Wavemeter (MaSWave) Based on a Short Section of Multimode Fiber as the Dispersive Element

**DOI:** 10.3390/s23104574

**Published:** 2023-05-09

**Authors:** Ogbole C. Inalegwu, Rex E. Gerald II, Jie Huang

**Affiliations:** Department of Electrical and Computer Engineering, Missouri University of Science and Technology, Rolla, MO 65409-0040, USA

**Keywords:** wavelength, multimode fiber (MMF), machine learning, convolutional neural network (CNN), specklegram, speckle patterns, charge-coupled device (CCD) camera

## Abstract

Wavemeters are very important for precise and accurate measurements of both pulses and continuous-wave optical sources. Conventional wavemeters employ gratings, prisms, and other wavelength-sensitive devices in their design. Here, we report a simple and low-cost wavemeter based on a section of multimode fiber (MMF). The concept is to correlate the multimodal interference pattern (i.e., speckle patterns or specklegrams) at the end face of an MMF with the wavelength of the input light source. Through a series of experiments, specklegrams from the end face of an MMF as captured by a CCD camera (acting as a low-cost interrogation unit) were analyzed using a convolutional neural network (CNN) model. The developed machine learning specklegram wavemeter (MaSWave) can accurately map specklegrams of wavelengths up to 1 pm resolution when employing a 0.1 m long MMF. Moreover, the CNN was trained with several categories of image datasets (from 10 nm to 1 pm wavelength shifts). In addition, analysis for different step-index and graded-index MMF types was carried out. The work shows how further robustness to the effects of environmental changes (mainly vibrations and temperature changes) can be achieved at the expense of decreased wavelength shift resolution, by employing a shorter length MMF section (e.g., 0.02 m long MMF). In summary, this work demonstrates how a machine learning model can be used for the analysis of specklegrams in the design of a wavemeter.

## 1. Introduction

Optical fibers enable sensing via the unique and reproducible modification of light rays propagated from one end of the fiber to the other. These rays have characteristic trajectories, and at the receiving end, a data acquisition system can be used to analyze the received signal for specific optical information. Example information of interest from optical signals could be the changes in the transmitted ray properties, such as the wavelength, velocity, and intensity of a light signal for a variety of measurement applications [1,2,3]. A wavelength meter or wavemeter is needed to measure the changes in wavelength for a variety of scientific research and industrial applications [4]. The analysis of wavelength in particular is critical for high-precision applications in atomic physics, metrology, and spectrometry [5,6]. Some of the traditional methods employed in the design of wavemeters are gratings [7,8], optical heterodyne systems [9,10,11], and Fabry–Perot interferometry systems [12,13]. Although these approaches have been around for decades, they still have some limitations, which are as follows: they require ultra-high free space alignment, they are bulky, and they are very expensive, with the cost and size of the interrogation system being the most prominent. Consequently, these drawbacks have led to research focused on the use of optical fibers. The miniature size and low cost of optical fibers can be exploited for replacing the bulky and expensive dispersive elements currently used in traditional wavelength meters. In [14], the authors used an MMF and a charge-coupled device (CCD) camera to analyze the multimodal interference pattern (i.e., the speckle patterns or specklegrams) captured by the camera, and they demonstrated that an MMF and a CCD camera can be set up to function as a spectrometer. Their study showed that the traveling rays of incident light along the core of an MMF interact in such a way as to produce a signature specklegram for a given wavelength, so long as the environmental conditions remain constant (i.e., there existed a one-to-one mapping between a wavelength and the corresponding specklegram for a fixed setup). The concept of using an MMF and a CCD camera has been recently deployed in several sensing applications requiring the analysis of speckle patterns [15]. In [16], the authors adopted the Talbot effect and a tone parameter extraction technique to design a high-precision (<10 pm uncertainty) wavemeter by investigating a sampled Talbot interferogram. Moreover, due to the material properties of optical fibers, they are extensively used to achieve compact and robust designs for a variety of applications [17,18]. Moreover, an application of MMF for spectral analysis was demonstrated in [19], where an MMF was used as a dispersive element; from the analysis of the captured speckle pattern, the authors were able to recover the spectral content of the transmitted signal by using calibration data. In addition, refs. [20,21] showed how this apparatus design—using an MMF and a CCD camera—can be used to reconstruct speckle patterns for an all-fiber spectrometer by analyzing the transition matrix from the changes in intensity produced by the different wavelengths. Further, in [22], a broadband MMF spectrometer was designed from the analysis of speckle patterns from five MMFs. A wavelength-dependent multiplexer (WDM) was used for the experiment. In the end, the authors developed an algorithm that could reconstruct the spectrum from captured speckle patterns with a reconstruction error (ε) of 0.003 within a 100 nm bandwidth. Furthermore, employing speckle pattern analysis using a stabilized diode laser source, the authors of [23] realized attometer resolution using the principal component analysis (PCA) method on time series speckle image data. The same authors also used PCA to separate femtometer-resolved multiple independent wavelength measurements with a 0.2 fm precision from a 1 m long step-index MMF [24]. In recent years, PCA speckle pattern analysis has particularly yielded higher scales of wavelength resolution beyond the traditional methods mentioned earlier [25,26,27]. Hence, it is evident that the concept of speckle pattern analysis has led to a variety of breakthroughs in optical sensing for the development of fiber specklegram sensors (FSS), some of which have been applied to chemical sensing [28], force myography sensing [29], modal power distribution sensing [30], temperature and weight sensing [31], fluid properties sensing [32], and mechanical stimuli sensing [33]. All these research examples support the fact that MMFs exhibit characteristic behavior which can be exploited for use in wavelength meter development [34].

On a related note, the advancement in computing power has led to the use of machine learning as a powerful tool for imaging and computer vision analysis. Machine learning has been applied in data analysis for the development of optical sensors such as the design of a fiber directional position sensor [35], the design of a multifunctional optical spectrum analyzer [36], the design of a fiber optic embedded smart helmet [37], the identification of materials through spectral analysis [38], holographic microscopy [39], and the reconstruction of handwritten digits [40]. Some other applications include structural health monitoring [41], digital holography [39], and mechanical measurements [42]. A major reason for these wide applications of machine learning is that the machine learning models can learn the underlying relationship between a set of data input(s) and the corresponding output(s) to substantially minute detail. Moreover, in most cases, machine learning models can process large amounts of data better than traditional statistical signal processing [43].

Although there have been several reports aimed at the development of fiber optic wavemeters, as mentioned above, only a few of these works are centered on improving the fiber optic wavemeter’s immunity to environmental stability. The main limitation of most of the existing wavemeters is that they are not simple to operate, as they require ultra-high free space alignment and are bulky; they are also quite expensive (above USD 7000 in purchase cost) mainly due to the costly interrogation units being employed. In this paper, a simple-to-operate (not requiring ultra-high free space alignment and with a lightweight apparatus) machine learning specklegram wavemeter (MaSWave) which uses a low-cost CCD camera (a Xeva 12,784 CCD camera that costs USD 1500) as the interrogator is reported. The MaSWave can operate over the entire bandwidth of a Hewlett Packard 8168F tunable laser (1440–1590 nm). In addition, by adopting the low-cost CCD camera for signal interrogation, and an optical fiber setup as the dispersive element, the cost of the MaSWave was reduced by thousands of dollars in comparison with existing commercial wavemeters. Moreover, the MaSWave was able to resolve the wavelength output over the range of the tunable laser source with a step size of 1 pm. The trained model classifier was employed on wavelength step sizes of 10 nm, 1 nm, and down to 1 pm, within the 1440–1590 nm operating range of the tunable laser. The training of the model did not use every possible wavelength output from the tunable laser source for training because that would have resulted in millions of training sets. In addition, the analyses of the specklegrams collected using the CCD camera for the conducted experiments show the adaptability of machine learning in this field of study. The work herein also demonstrates that by employing machine learning, researchers can overcome the usual trade-off between the spectral resolution and the operation bandwidth (i.e., the capability to classify all wavelengths down to 1 pm wavelength shift for the entire bandwidth of the tunable laser source). The MaSWave design was able to exploit the optical signature in every specklegram because there exists a signature optical fiber dispersion-medium specklegram for every wavelength of light generated by a light source. In summary, we show how the closest possible wavelength shift (1 pm) outputted by the Hewlett Packard 8168F tunable laser source can be identified. Notably, the data processing did not require a high-end computing system. The analysis was performed using a Dell Core i7 PC, 16G RAM, running Windows 10, to demonstrate that a high-end computing system was not required for the implementation of the MaSWave.

## 2. Methodology

### 2.1. Sensing Principle

Specklegrams and speckle patterns are the terms used to describe the characteristic spatial arrangements of bright and dark spots (speckles) captured at the end face of an MMF (acting as an optical dispersive element) by a CCD camera. Specklegrams occur because of the interferences between the propagating modes in the core of an MMF. The interferences can be either constructive (resulting in bright spots) or destructive (resulting in dark spots). Importantly, the observed speckle patterns exhibit variations in pixel intensity that are spatially distributed. Therefore, traditional analysis of the nature of specklegrams typically adopts statistical signal processing tools [42,44]. For a specklegram with a sum of *N* independent speckle intensities given as *I*, and the mean value of the total intensities 〈I〉, the probability density function p(I) can be expressed as [42]:(1)$$p\left(I\right)=\frac{I}{\langle I\rangle }\exp\left(\frac{-I}{\langle I\rangle }\right)\;I\;\geq 0\;$$

An approximation of the total number of guided modes resulting in the spatially distributed bright and dark speckles, each with phase information distributed over (−π, π), is expressed as [20]:(2)$$N_{spots}\approx \frac{\pi^{2}NA^{2}d_{co}{}^{2}}{2\lambda^{2}}$$
where NA and $d_{co}$ are the numerical aperture and the core diameter of the MMF, respectively, and λ is the propagation wavelength of the transmitted light signal from an optical source. From Equation (2), it is numerically possible to differentiate the $N_{spots}$ of different wavelengths for speckle patterns of wavelength values that are far apart; however, it becomes very complicated to do so for closely spaced wavelengths where the resultant $N_{spots}$ values are not significantly different. Moreover, a shift in the wavelength results in a corresponding change in each guided mode propagation constant ($\beta_{m}$), where $\beta_{m}$ is for the mth guided mode; consequently, the change in the propagation constant results in a change in the phase delay ($\emptyset_{m}$) for the mth mode as well, thus leading to the relationship between λ and $\beta_{m}$ represented by [19]:(3)$$\Delta \beta_{m}=\beta_{m}\left(\lambda +\Delta \lambda \right)-\beta_{m}\left(\lambda \right)$$
and the corresponding relationship between $\beta_{m}$, $\emptyset_{m},$ and the length of the fiber (L) is expressed as [19]:(4)$$\Delta \emptyset_{m}=\Delta \beta_{m}L$$

Since the bright and dark spots will have different pixel intensities, sizes, and spatial distributions at different wavelengths, the task of accurately mapping a specklegram to its signature wavelength becomes challenging. While traditional statistical approaches can be used to obtain the $N_{spots}$ value for wavelength determination, the traditional statistical methods are limited by the wavelength step sizes (i.e., the shift in wavelength), and the evolving intensity, size, and spatial distribution of the speckles for every wavelength shift. For this reason, a machine learning algorithm was determined to be best suited for the task of predicting the wavelength from the input specklegrams. Further, the decorrelation in speckle patterns is easier to resolve in longer fibers, as expressed in Equation (4). Therefore, in [19], we see that a 1 m long fiber achieved a spectral resolution of 0.15 nm while a 5 m long fiber showed a spectral resolution performance of 0.03 nm using the transmission matrix technique for the analysis; both cases were over a bandwidth of 25 nm and 5 nm, respectively. Additionally, in [14], the authors used a 100 m long MMF to achieve 1 pm resolution with a similar setup. However, one challenge with increasing the length of the fiber is that it makes it more susceptible to environmental influences, particularly vibrations and temperature changes. In this paper, the first experiment was conducted with an initial 16 m long MMF to show the susceptibility to environmental disturbances. Later, the length of the MMF was scaled down to 0.1 m by splicing with a single-mode fiber (SMF) to improve the stability—by reducing the length of the MMF section, the apparatus is made more robust to the effects of environmental disturbances. The splicing had to be performed to compensate for the length (i.e., to function as a fiber extension for the light profile transmitted from the laser source to the MMF dispersive element) needed to connect one end of the fiber to the tunable laser, as illustrated in Figure 1.

The Hewlett Packard 8168F tunable laser served as the optical source for the transmitted light of a specific wavelength (λ) through the MMF dispersive element, while the CCD camera was used to capture the speckle pattern at the end face of the MMF. The captured speckle pattern was then saved on the PC for post-processing. In the end, the 0.1 m long MMF, with a core diameter of 105 µm and NA of 0.22, was found to be immune to the vibrations within the lab space. Interestingly, for both the 16 m and 0.1 m MMF lengths, we were able to show that it is possible to detect the change in speckle patterns for up to 1 pm of wavelength shift. However, the 0.1 m long MMF was substantially more immune to vibration and temperature changes from the surrounding environment, unlike the 16 m long MMF. Further, a 0.02 m long MMF with a similar structure (i.e., an SMF spliced to an MMF) was shown to be capable of resolving up to a 1 nm shift in wavelength. 

### 2.2. Data Acquisition

Figure 1 shows one end of the MMF connected to the tunable laser source, and its distal end securely placed before the CCD camera 0.05 m away from the camera lens. The image at the end face of the MMF was captured by the CCD camera. All data reported in this work were acquired at the Lightwave Technology Lab (LTL), Missouri S&T, with the experimental setup as depicted in Figure 1. A total of 9000–16,000 specklegram images were used for each training session (depending on the number of classes considered in the given training session); in the course of the entire work, over 100,000 specklegram images were captured for the different wavelength classification tasks (i.e., from the 10 nm to 1 pm wavelength shift cases). The first round of image acquisition was carried out for the entire operating wavelength range of the tunable laser source at wavelength shifts of 10 nm. After several attempts at training the model with different image training sizes (i.e., the number of training images per task), 900 specklegram images per class were found to be an optimal size; hence, 900 specklegram images were further used for each class to maintain a balanced dataset across all classes (i.e., the wavelength shifts) considered. Notably, it is recommended to provide a machine learning model with multiple examples of a given class of an item to enable the model to learn subtle differences in features between images in the same class that are not noticeable by visual inspection, and thus help the model to improve on its generalization ability. Afterward, smaller wavelength shifts of 1 nm, 0.1 nm, 0.01 nm, and finally 0.001 nm (1 pm) step sizes were captured—1 pm was the smallest wavelength step size the tunable laser could achieve. For the wavelength prediction task with smaller step sizes, the captured images were near 1500 nm; this operational wavelength was used because it is at the middle of the operational range of the tunable laser source, and a 1 pm step size would result in millions of speckle images to analyze if adopted for the entire operational range of the equipment. However, to test if the machine learning model can accurately generalize across the entire wavelength range of the tunable laser light source, similar data were collected at 1440–1449 nm and 1580–1590 nm.

Upon examining Figure 2, it is evident that the closer the wavelengths for adjacent specklegram images, the more identical the images appear. In the course of the experiment, it was noticed that the machine learning model employed in the classification task was able to detect subtle changes that existed in some of the specklegram images within the same class, but these differences were not significant enough to lead to misclassifications for the wavelength-dependent specklegrams. All specklegram images shown were captured at room temperature.

### 2.3. CNN Model

A CNN machine learning model was found to be appropriate for the specklegram-to-wavelength prediction task for the MaSWave design, especially since we are dealing with an image dataset. Recent research works have consistently shown successful outcomes with CNNs on FSS applications [27]. The specific model used for the task (termed CNN_Model in this report) was able to identify the underlying correlations and accurately map a specklegram to its signature wavelength. Table 1 summarizes the structure of the CNN_Model.

The CNN_Model as depicted in Table 1 has six layers in all. The first two layers are a series of convolution and pooling layers with a kernel size of 3 × 3 and pool size of 2 × 2, respectively. The convolution unit on both layers uses the rectified linear unit (ReLU) activation function. The ReLU function does not update all neurons at the same time, and therefore the positive portion is updated more rapidly during training, which helps to speed up the process. Next is the flatten layer, which converts the image data to a 1-dimensional array feature vector. The flatten layer is followed by a dense layer that performs the matrix-vector multiplication and adopts a ReLU activation function as well. After the first dense layer (Dense_1) is another dense layer (Dense_2) with a dropout function to prevent the model from being overfitted. Finally, there is a third dense layer (Dense_3) with a Softmax activation function for the classification process, as depicted in Figure 3.

The CNN_Model was developed on a Jupyter notebook within the Anaconda software package. The Anaconda package is an open-source platform for Python data science computation, and the design utilized the Keras API running on Windows 10 and a TensorFlow machine learning library. The Anaconda package was chosen for this project because it is user-friendly and comes with many other development tools for data science analysis.

## 3. Results and Discussion

The first training set was on speckle patterns for 10 nm sequential wavelength step sizes. Major tuning of the model’s hyperparameters was carried out on the CNN_Model during the first training case. After adjustment of the parameters, a model with a learning rate = 0.001, batch size = 20, and beta value = 0.9 was shown to achieve an accuracy of 100%. The dataset had 9000 specklegram images (captured by the CCD camera from the experiment conducted) of which 8000 images were used for training (with validation split = 0.15), and the remaining 1000 images were reserved to test the model. This initial model was then deployed for the next set of cases with wavelength step sizes of 1 nm, 0.1 nm, 0.01 nm, and 0.001 nm (1 pm). Similarly, for each of these subsequent cases, 8000 specklegram images were also used for the training, and 1000 were used to test the model.

### 3.1. Specklegram Correlation

First, the images were converted to arrays with pixel values in the range [0, 255] before being fed to the network. The image shown in Figure 4 was from the end face of the 0.1 m long MMF with a 105 µm core diameter. The two-dimensional (2D) image obtained by the CCD camera allows the CNN_Model to project the pixel intensity distribution onto a 2D plane (*x*,*y* coordinates for processing), as captured in Figure 4.

The pixel values correspond to the pixel intensity values for the spatially distributed modes along the *x*,*y* coordinates (i.e., *I*(*x*,*y*) for each specklegram). A high pixel value, for example, 240, indicates an in-phase interference between several modes, and a value of 10 indicates a high degree of out-of-phase interference between the modes. Thus, the degree of pixel intensity ranges from 0 to 255, with 0 as the darkest possible spot and 255 as the brightest possible spot. To find the correlation between two specklegram images after a change in the wavelength, the correlation coefficient C(Δλ) was used, and it is expressed as [45]:(5)$$C\left(\Delta \lambda \right)=\frac{\langle I_{0}\left(x,y\right)I_{1}\left(x,y\right)\rangle -\langle I_{0}\left(x,y\right)\rangle \langle I_{1}\left(x,y\right)\rangle }{\sqrt{{\left(\langle I_{0}^{2}(x,y)\rangle -\langle I_{0}{(x,y)\rangle }^{2}\right)}^{2}\;{\left(\langle I_{1}^{2}\left(x,y\right)\rangle -\langle I_{1}{(x,y)\rangle }^{2}\right)}^{2}}}$$
where 〈…〉 represents the mean averaging over the spatial coordinate x and y for a given wavelength-dependent specklegram. $I_{0}\left(x,y\right)$ is the spatial pixel intensity of the reference specklegram, and $I_{1}\left(x,y\right)$ is the spatial pixel intensity of a specklegram after a change in the wavelength (Δλ). If the correlation coefficient C(Δλ) is approximately unity (C(Δλ) ≈ 1), the specklegram images are nearly identical. The specklegrams are nearly completely uncorrelated for C(Δλ) ≈ 0. The success of the specklegram correlation approach depends on the degree of decorrelation that the machine learning model can detect between transitions from one wavelength to another. The correct prediction is influenced by the weight updates during the training of the network, from using the correct hyperparameters and transfer functions presented by the CNN_Model. Finally, the correlation coefficient computation is carried out for all images presented to the network during training and test analysis.

After several training iterations (with modifications to hyperparameters), an accuracy of 100% was still achieved with the CNN_Model. A prediction accuracy of 100% means that for all images of the different classes presented to the model, the network correctly identified the image class for each of the test images. From the analysis, all the actual predictions were not exactly 100% in probability values—probability values ranging from 98–99.99% for the correct classes were obtained. However, the algorithm compared the probability values for all classes (i.e., all possible outcomes) and selected the class with the maximum probability value as the predicted class. For example, if the probability that a given test image shares features that are 98% similar to those of class ‘A’, 1.5% similarity to class ‘B’, and 0.5% similar to class ‘C’ in the training outcome, the model would classify the specklegram image as a class ‘A’ image. The final selection of a single class with the maximum probability value is achieved through the operation of the Softmax activation function used by the CNN_Model. The operation of the Softmax activation function at the classification layer enables the model to come up with an accurate prediction 100% of the time for the classification task. Table 2 shows the result of the CNN_Model adopted for the classification task. In addition, VGG16 and RESNET50, both award-winning models for image classification challenges, were also tried on the specklegram dataset—to test how well these pre-trained models will perform on the experimental dataset without any adjustments of parameters for the 1 pm wavelength shifts. The networks were all fed with a balanced dataset (i.e., the same number of images for each class). If a balance in the experimental dataset is not maintained, each classifier is very likely to be biased in favor of the more represented class.

To further test the generalization of the developed model and the repeatability of the approach, a group of untrained images taken at a later time was analyzed. This also yielded a 100% prediction accuracy for the final predicted class. However, the similarity between specklegram images of the same wavelength dropped to an average of 96% after a 24 h period. The reason for the drop in similarity is most likely due to the changes in the ambient conditions in the lab over a 24 h period causing the MMF dispersion element to slightly change physical characteristics. The 4% dissimilarity in the newly captured specklegrams, when compared with the control specklegrams earlier captured, was not significant enough to lead to misclassification by the CNN_Model. Interestingly, the dissimilarity was not even noticeable to the naked eye, but the model was able to spot the differences. 

The results presented thus far indicate that specklegram-based analysis is effective for accurately classifying narrowly spaced wavelengths using machine learning. Although it may take some time to develop an effective and efficient model, once a working model has been developed it becomes easy to adapt, making the approach scalable. For example, Table 2 also includes the results from two other models, VGG16 and RESNET50, both off-the-shelf models (i.e., they have both been pre-trained on a different dataset), and they both performed well when applied on the specklegram dataset without any modifications to the hyperparameters. Some preprocessing of the dataset was performed on the image dimensions, where all images were set to 256 × 256 pixel dimensions for uniformity. In addition, the dark background was cropped in some of the machine learning analyses, but this did not significantly affect the training and test results. The datasets fed for training were shuffled, and each image had its unique identifier as a ground truth label.

### 3.2. Improving Robustness to Environmental Changes

A major challenge associated with the analysis of specklegram-based sensors is their susceptibility to environmental influences, particularly temperature and mechanical vibrations. In this study, the analysis using a 16 m long MMF showed excellent results for the classification of the wavelength-dependent specklegrams. However, the 16 m long MMF was highly susceptible to temperature variations. A temperature change of 1 °C experienced by the MMF was enough to bring about noticeable changes in the specklegram, which negatively affected the prediction accuracy of the MaSWave. To overcome this challenge, a much shorter MMF section (0.1 m long, 105 µm core diameter, and 0.22 NA step-index MMF) spliced to an SMF was utilized. The longer-length SMF spliced to the shorter-length MMF structure made the design less susceptible to changes in temperature and vibration. An oven was used to monitor the effect of temperature fluctuations on the MMF dispersive element by heating the MMF section (i.e., the sensing section). A temperature change on the sensing section of the fiber is expected to result in changes in the observed specklegram, but the SMF–MMF structure configuration showed indistinguishable specklegram images for variations in temperature of up to ±3 °C, for an observation period of 2 h. Furthermore, the 0.1 m long MMF setup was monitored for 24 h under the ±3 °C temperature variations, and it showed an approximately 4% variation in the similarity of the new specklegram when compared to the control specklegram. In the case of the 16 m long MMF, the sensing section showed an approximately 35% variation for only a ±1 °C variation in temperature, which is a poor and unreliable outcome, especially for a wavemeter design. Additionally, to test the effect of external vibrations on the MaSWave, the 16 m long MMF set-up was intentionally made to vibrate by tapping the optical table. The outcome from the forced vibration was recorded, and the result showed that the individual speckle’s spatial orientation (for a case where all other conditions remained the same) would always return to its initial spatial orientation 3–5 s after the effect of the applied force was removed. For a firm structure, a fiber holder with a 300 µm diameter groove was used to secure the MMF dispersive element placed in the groove (see Figure 1). Therefore, the MaSWave design with a firmly fixed 0.1 m long MMF dispersive element secured within a groove on the fiber holder was able to significantly address the issue of susceptibility to vibrations from the surrounding environment and was found to be stable with fluctuations in temperature of up to ±3 °C.

### 3.3. Comparing Results for Different MMFs

In the analysis of different MMF dispersive elements for the design of the MaSWave, fiber attributes such as the length of the MMF, the core diameter of the MMF, and the characteristic index (step index or graded index) of the MMF were investigated. First, a 16 m long step-index MMF dispersive element with a core diameter of 400 µm and NA of 0.22 was implemented. After noticing that the length made it highly susceptible to changes in temperature and vibrations from the surrounding environment, we tested a significantly shorter MMF. Thus, the next major implementation was carried out using a 0.1 m long step-index MMF with a core diameter of 105 µm and NA of 0.22. Although both the 16 m long step-index MMF (core diameter of 400 µm and NA of 0.22) and the 0.1 m long step-index MMF (core diameter of 105 µm and NA of 0.22) exhibited a 100% accuracy in the test results for a 1 pm wavelength shift, the 0.1 m long MMF was significantly more immune to vibration and temperature changes. The third test was conducted with a 0.1 m long step-index MMF with a core diameter of 105 µm and a reduced NA of 0.11; the measured accuracy showed 81.77% in this case, and the observed stability was found to be similar to that of the 0.22 NA MMF’s. The first three analyses provided an indication of the effects of reduced length, reduced core diameter, and a reduced NA for a step-index MMF acting as the dispersive element for the MaSWave design, as shown in Table 3. Further, an additional 0.1 m long graded-index MMF was tested (core diameter of 62.5 µm and NA of 0.2), yielding an accuracy of 98.89%, and a vibration stability comparable to those of the previous 0.1 m long MMFs. Therefore, an MMF length of 0.1 m with an NA of 0.22, was demonstrated to be most suitable for surviving the vibrations within the lab space and could resolve a 1 pm wavelength shift. Table 3 tabulates the results from the MMFs used, beginning with the outcome from the first experiment with the 16 m long step-index MMF. The experiments in this section were conducted on the same optical stage and under the same environmental conditions to the best of our abilities. The step size was set to 1 pm for all instances presented in Table 3.

The results from Table 3 show that decreasing the numerical aperture makes it more difficult for the model to correctly classify each image. The reduced accuracy can be said to be a result of the reduced multimodal interference leading to lesser decorrelation between transitions in wavelength shift values. In addition, the larger the core size of the MMF used, the more speckle elements (the characteristic bright and dark spots) contained in each image were observed; both the increase in the core size and NA play a role in the level of observable multimodal interferences within the 0.1 m length of the MMF’s section, leading to a better overall result for the 105 µm, 0.22 NA step-index MMF finally adopted for the MaSWave design. In general, an increase in the length, the core size, and the NA of the MMF dispersive elements results in varying degrees of decorrelation in the specklegram, as expressed earlier in Equations (2) and (4). The same model was used for all cases considered. For each case, training was performed on specklegrams from the MMF under consideration. The re-training was necessary to provide a fair basis for comparison of the test results.

### 3.4. Sacrificing Resolution for Increased Robustness

Finally, to probe the limit of the length for the MMF section for the proposed design, the length of the step-index MMF section with a core diameter of 105 µm was reduced to 0.02 m, and the data were collected and tested for specklegram-to-wavelength classification accuracy. For this instance, the CNN_Model could only effectively classify specklegrams that were separated by a 1 nm wavelength shift, because the model could not detect any change in the specklegram images for a 1 pm shift. Nonetheless, the 0.02 m MMF spliced to the SMF showed specklegrams that were unchanging even for temperature variations of up to ±5 °C and was more insensitive to mechanical vibrations when compared to the 0.1 m long MMF.

## 4. Conclusions

This work presents a machine-learning-based specklegram wavemeter (MaSWave) that uses a short section of MMF as a dispersive element that is robust to temperature variations and mechanical vibrations. The work demonstrates that it is possible to accurately classify all possible resolved wavelength-dependent speckle patterns from the section of an MMF acting as the dispersive element. A CCD camera functioning as a low-cost interrogator was used to capture the specklegrams at the end face of the MMF, thus replacing the expensive interrogation systems commonly employed. The dispersive element (i.e., the MMF) was shown to be affected by vibrations and temperature variations, which in turn affected the specklegram output. The study demonstrated that the use of a short section of MMF (0.1 m long) effectively mitigated these challenges while still achieving 100% accuracy for a 1 pm wavelength shift. Further, using an optical table and fiber holders for the experiment made it possible for the stage conditions to be maintained all through the various data collection periods. Hence, for the deployment of the MaSWave outside of the laboratory space, the design requires robust packaging suitable for isolating the MMF from vibrations and insulating the MMF from temperature variations beyond ±3 °C (an in-built temperature controller can be used to maintain the operating temperature). In summary, the MaSWave design is cost-effective and simple to implement and operate, since it uses only a section of MMF dispersive element that is lightweight and does not require ultra-high free space alignment, which are both major drawbacks of traditional wavemeters. Moreover, all the data post-processing was carried out on a core i7 Dell PC with 16G RAM, buttressing the fact that the developed MaSWave introduces an affordable approach for wavelength metering. The transfer learning ability of machine learning models makes the MaSWave design easily deployable and adaptable to a wide range of wavemeter applications. Future work will consider the adoption of the MMF and CCD camera combination described herein for wavelength prediction with a regression model and for developing other fiber specklegram sensors (FSS). In addition, we surmise that the features of the speckles are a function of the cross-section of the core of the MMF dispersive element. Other cross-section geometries (e.g., triangular-shaped or square-shaped MMF cores) can be considered for investigating improved sensitivity in FSS applications. We strongly believe that non-cylindrical core cross-sections will yield greater decorrelations of the speckles, especially suited for sensing angular displacements in a fixed-fiber position (useful for metering torsional forces), which would probably go unnoticed in a cylindrical-core MMF in angular displacement applications, thus improving the sensing resolution in FSS applications.

## Figures and Tables

**Figure 1 sensors-23-04574-f001:**
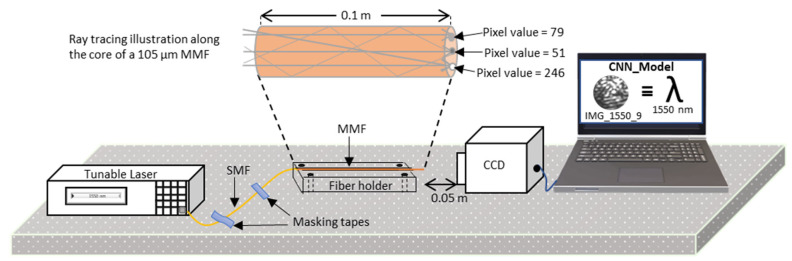
Schematic drawing of the experimental setup for a wavemeter that takes advantage of the specklegram-to-wavelength signature mapping of incident light. A Hewlett Packard 8168F tunable laser was used as the light source that couples the light to an SMF, which is spliced to an MMF. The 0.1 m long MMF, with a core diameter of 105 µm and NA of 0.22, serves as the dispersive element where the multi-modal interferences of the propagating modes are revealed. A Xeva 12,784 CCD camera is used to capture the interference image (specklegram) at the end face of the MMF. The specklegrams were processed by a PC for wavelength prediction using a CNN model. An illustration of several light ray paths is used to show how the interaction between the propagating modes results in different pixel intensity values at the fiber’s end face. Each captured image was saved with a unique file name (e.g., IMG_1550_9) that served as a ground truth label, and the predicted wavelength is expressed as a numerical value (e.g., 1550 nm).

**Figure 2 sensors-23-04574-f002:**
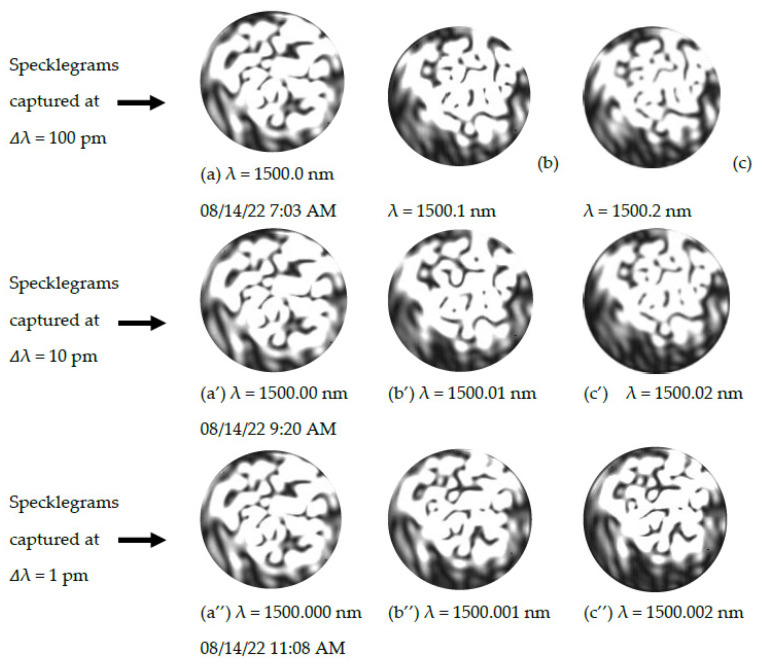
A tabular display of specklegrams (with the uniform black background cropped) captured at different wavelengths and times using a CCD camera. The images reveal the evolution of the specklegrams as a result of the changes in the wavelength of the tunable laser light source. It is difficult to visually identify changes in the specklegrams recorded for the same light input wavelength at different times. Images (**a**), (**a′**), and (**a″**) are the control images taken approximately 2 h apart; they appear to be identical to the naked eye, as shown in the figure. Top row: Images (**a**–**c**) are specklegrams for a wavelength shift of 100 pm from an initial wavelength of 1500 nm. Middle row: images (**a′**–**c′**) are specklegrams for 10 pm shifts in the wavelength, also initiated at 1500 nm. Bottom row: images (**a″**–**c″**) represent the specklegram images for a wavelength shift of 1 pm, with reference wavelength (λ) = 1500 nm.

**Figure 3 sensors-23-04574-f003:**
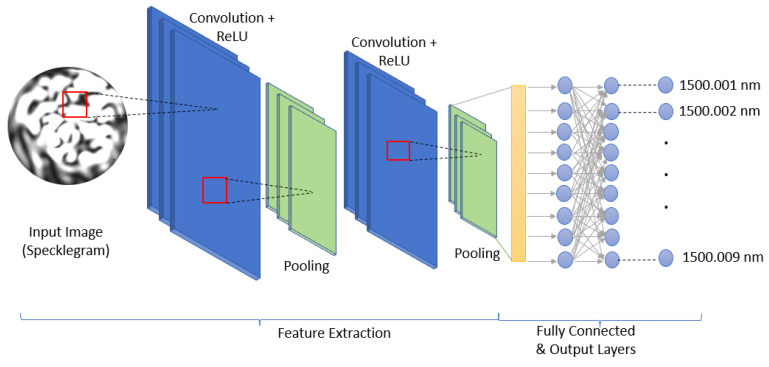
CNN model showing the input specklegram image and the different layers of the network: convolution layer, pooling layer, flatten layer, fully connected layer, and the output layer. The output illustrates the classes considered in the wavelength prediction task.

**Figure 4 sensors-23-04574-f004:**
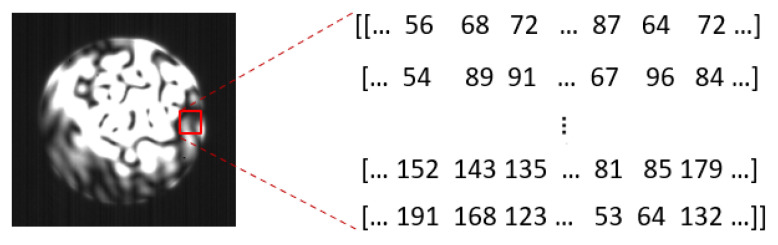
A depiction of how the CNN_Model analyzes specklegram images captured at the end face of an MMF. The CNN_Model considers each specklegram as a 256 × 256 array of pixel values with pixel intensities *I*(*x*,*y*) across the entire image. The dark spots are regions of destructive interference of the propagating modes (resulting in lower pixel intensity values) while the brighter regions are an indication of constructive interference between the modes (resulting in higher pixel intensity values).

**Table 1 sensors-23-04574-t001:** The CNN_Model.

Layers	Type	Kernel/Pool Sizes	Activation Function
1	Convolution2D_1 Maxpooling2D_1	Kernel size = 3 × 3 Pool size = 2 × 2	ReLU -
2	Convolution2D_2 Maxpooling2D_2	Kernel size = 3 × 3 Pool size = 2 × 2	ReLU -
3	Flatten	-	-
4	Dense_1	-	ReLU
5	Dense_2(with Dropout)		-
6	Dense_3		Softmax

**Table 2 sensors-23-04574-t002:** Training and test results for the different models.

Step Size	Model	Number of Images		Accuracy (%)	
		Train	Test	Train	Test
1 pm	CNN_Model	8000	1000	100	100
1 pm	VGG16	8000	1000	96.1	74.5
1 pm	RESNET50	8000	1000	91.2	67.4

**Table 3 sensors-23-04574-t003:** Comparing the classification accuracy for different MMFs.

MMF Type	Length of the MMF (in m)	Core Diameter of MMF (in µm)	Numerical Aperture (NA)	Classification Accuracy (%)
Step-index	16	400	0.22	100
Step-index	0.1	105	0.22	100
Step-index	0.1	105	0.11	81.77
Graded-index	0.1	62.5	0.2	98.89

## Data Availability

All data were collected at the Lightwave Technology Lab Missouri S&T and can be made available on request.

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
