# Peer review of "A Machine Learning Specklegram Wavemeter (MaSWave) Based on a Short Section of Multimode Fiber as the Dispersive Element"

_sensors, 2023, doi:10.3390/s23104574_

Round 1

Reviewer 1 Report

The article has written well but the author needs to clarify the below comments

1. The author has not defined the source of the data taken and to create the machine learning model what techniques or methods used is not clear.

2. In Section 2.3, author mentioned that CNN model is best among all the model, can author specify in what way your chosen CNN model is best in all the cases.

3. How or in what method the training and test data has taken, there are no clues about this.

4. Its interesting that the result of the accuracy for both training and testing has secured 100 percent. The author has not specified in which language the implementation has done to achieve this much percentage. Its requested to supply the coding part to verify that without RSME how the 100 percent attained.

5. There is no visualization on this part, its informed to add the visualization part in the article. Authors can refer this article https://doi.org/10.3390/electronics11244178  and include the visualization.

Author Response

Thank you.

Reviewer 2 Report

The manuscript reports on a wavemeter based on a multimode fiber and a CCD camera. However, all-fiber wavemeters and spectrometers aren't a new topic and are investigated for more than 10 years with the reported results surpassing those presented in the manuscript (ability to reconstruct the spectrum of the incoming light instead of measuring the position of a single wavelength as well as higher wavelength resolutions). For instance, a very interesting paper [https://doi.org/10.1364/OL.388960] was missed in the references, where an ability to resolve several wavelengths with fm-level accuracy was reported.

However, the authors investigate important questions about the robustness of their wavemeter, which makes it potentially possible to accept the manuscript for publication. Below I list the comments that must be addressed by the authors.

1. First of all, it is unclear why did the authors choose to train the model in a classification mode, when the task of wavelength estimation requires a continuous output that can be provided by a regression model.

2. Moreover, it is unclear how the speckle correlation is used in model training and how the final accuracy is related to it (if is at all).

3. It is also omitted in the text, but is a crucial, whether the authors used the same model for all multimode fibers or did they have to re-train it for different fibers.

4. It is unclear how it was possible to achieve higher wavemeter accuracy for graded-index fiber than for a step-index fiber with NA=0.11 (graded-index fiber still must have had much higher bandwidth and hence, lower dispersion, also, typically, much lower number of modes are launched in a graded-index fiber than in step-index fiber using SMF).

5. It is better to state explicitly what was the wavelength error in case of heating the 0.1 m fiber. This way, it will be easier to conclude is there any sense in using shorter fiber.

6. When the authors state that the wavemeter based on 0.02-m MMF "was more insensitive to mechanical vibrations" than the wavemeter based on 0.1-m MMF? they need to be more specific (what was the wavelength error in these two cases).

Author Response

Thank you.

Reviewer 3 Report

It is novel that a machine learning-based specklegram wavemeter (MaSWave) is developed by use of a short section of MMF as a dispersive element, which is robust to temperature variations and mechanical vibrations. The theoretical and experimental researches are effectively ensured that the accuracy of the MaSWave is high enough to be used in practice. And the machine learning model is induced in specklegram traning make the work novel in present. 

Author Response

Thank you.

Round 2

Reviewer 1 Report

This revision has done well. Accept